# Association between Polypharmacy and Cardiovascular Autonomic Function among Elderly Patients in an Urban Municipality Area of Kolkata, India: A Record-Based Cross-Sectional Study

**DOI:** 10.3390/geriatrics7060136

**Published:** 2022-11-29

**Authors:** Shambo Samrat Samajdar, Saibal Das, Sougata Sarkar, Shatavisa Mukherjee, Ashish Pathak, Cecilia Stålsby Lundborg, Indranil Saha, Santanu Kumar Tripathi, Jyotirmoy Pal, Nandini Chatterjee, Shashank R Joshi

**Affiliations:** 1Department of Clinical and Experimental Pharmacology, Calcutta School of Tropical Medicine, Kolkata 700 073, India; 2Diabetes and Allergy-Asthma Therapeutics Specialty Clinic, Kolkata 700 014, India; 3Indian Council of Medical Research—Centre for Ageing and Mental Health, Kolkata 700 091, India; 4Department of Global Public Health, Karolinska Institutet, 171 77 Stockholm, Sweden; 5Department of Pediatrics, RD Gardi Medical College, Ujjain 456 006, India; 6Department of Pharmacology, Netaji Subhas Medical College and Hospital, Patna 801 106, India; 7Department of Medicine, RG Kar College and Hospital, Kolkata 700 004, India; 8Department of Medicine, Institute of Postgraduate Medical Education and Research and SSKM Hospital, Kolkata 700 020, India; 9Joshi Clinic, Mumbai 400 050, India

**Keywords:** autonomic function, cardiac, elderly, medication, polypharmacy

## Abstract

We assessed the association between polypharmacy and cardiovascular autonomic function among community-dwelling elderly patients having chronic diseases. Three hundred and twenty-one patients from an urban municipality area of Kolkata, India were studied in August 2022. The anticholinergic burden and cardiac autonomic function (Valsalva ratio, orthostatic hypotension, change in diastolic blood pressure after an isometric exercise, and heart rate variability during expiration and inspiration) were evaluated. Binary logistic regression analysis was performed to find out the association of polypharmacy and total anticholinergic burden with cardiac autonomic neuropathy. A total of 305 patients (age, 68.9 ± 3.4; 65.9% male) were included. Of these patients, 81 (26.6%) were on polypharmacy. Out of these 81 patients, 42 patients were on ninety-eight potential inappropriate medications. The anticholinergic burden and the proportion of patients with cardiac autonomic neuropathy were significantly higher among patients who were on polypharmacy than those who were not (8.1 ± 2.3 vs. 2.3 ± 0.9; *p* = 0.03 and 56.8% vs. 44.6%; *p* = 0.01). The presence of polypharmacy and a total anticholinergic burden of > 3 was significantly associated with cardiac autonomic neuropathy (aOR, 2.66; 95% CI, 0.91–3.98 and aOR, 2.51; 95% CI, 0.99–3.52, respectively). Thus, polypharmacy was significantly associated with cardiac autonomic neuropathy among community-dwelling elderly patients.

## 1. Introduction

Polypharmacy is the concurrent use of multiple medicines. Although there is no unanimous definition, polypharmacy is considered the routine use of five or more medicines [1]. Its prevalence is around 40–50% in high-income countries [2,3]; while in Indian studies, it is reported to vary from 2–93% [4,5]. Elderly (≥ 60 years of age) individuals are at a high risk to experience polypharmacy [6]. Various global studies have shown that two to nine medicines per day on average are taken by elderly individuals [7]. A study reported that 91% of patients aged 57–85 years took at least one medication [8]. As the global population is aging rapidly, polypharmacy has been recognized as an important medical concern considering the various risks it imposes [1].

Altered physiological status and declining organ functions in elderly patients lead to alterations in the pharmacokinetics of various medicines [9,10]. Furthermore, these changes vary across elderly patients with passing decades [11]. Polypharmacy often results in potential inappropriate medications, causing considerable harm [1,6,10]. Polypharmacy in elderly patients can lead to non-adherence to medications directly affecting treatment outcomes [12,13]. The risk of potential serious drug–drug interactions also increases with polypharmacy [14]. Polypharmacy, as an independent variable, has been linked to falls and increased hospital admissions in elderly individuals [15,16]. Furthermore, polypharmacy also increases the odds of cognitive and functional impairment, urinary retention, delirium, and mood disorders among elderly individuals [17,18]. Elderly patients and those on polypharmacy are particularly at risk of blood pressure impairment and orthostatic hypotension [19,20,21]. It has been found that pharmacotherapy is one of the main causes of orthostatic blood pressure impairment, leading to iatrogenic orthostatic hypotension [21].

The cardiac autonomic nervous system is impaired in elderly individuals [22], and polypharmacy may add to the risk [23]. A study conducted in Japan found a relationship between reduced heart rate variability, a marker of cardiac autonomic function, and polypharmacy among elderly patients after adjusting for the patients’ comorbidities [23]. However, the association between polypharmacy and comprehensive parameters of cardiovascular autonomic function in elderly patients has not been studied much in the Indian population. We aimed to assess the association between polypharmacy and cardiovascular autonomic function among elderly patients attending a Clinical Pharmacology health consultation clinic in an urban municipality area of Kolkata, India.

## 2. Materials and Methods

### 2.1. Study Design and Setting

This was a record-based cross-sectional study conducted at a community-based Clinical Pharmacology health consultation clinic for elderly patients in an urban municipality area of Kolkata, India in August 2022. 

### 2.2. Study Population

All elderly patients aged ≥60 years with chronic diseases (diseases of long duration and resulting from a combination of genetic, physiological, environmental, and behavioral factors [24]) who attended the health consultation clinic for medication review and reconciliation were included in the analysis. Elderly patients who were deemed physically unfit by the investigators to undergo cardiac autonomic function tests and those having cancer were excluded from the analysis. 

### 2.3. Baseline Assessment

Demographic characteristics (age and sex), present medical history (including the status of disease control at the time of enrolment), and medication history of the patients were recorded at baseline. Polypharmacy was considered if the patients were taking five or more oral allopathic medicines routinely [1]. The potential inappropriate medications among the patients who were on polypharmacy were identified using the STOPP/START (screening tool of older people’s prescriptions and screening tool to alert to right treatment) criteria [25]. The anticholinergic burden of the individual medicines was assessed on a score of 0–3, according to their anticholinergic effects. A score of 0 represented no anticholinergic effect, a score of 1 signified possible, and scores of 2 and 3 implied definite anticholinergic effects [26]. The total anticholinergic burden in each patient was calculated. 

### 2.4. Assessment of Cardiac Autonomic Function

Standard and uniform protocols were followed to assess cardiac autonomic function. The Valsalva ratio was calculated by a ratio of the maximum heart rate during the Valsalva maneuver (forced expiration against resistance for at least 7 s) over the minimum heart rate 30 s after the maneuver. A Valsalva ratio of <1.1 was considered abnormal [27,28]. Orthostatic hypotension was defined by a fall in systolic blood pressure of ≥20 mm Hg (≥30 mm Hg in patients with hypertension) and/or a fall in diastolic blood pressure of ≥10 mm Hg within 3 min of standing [28,29]. The rise in diastolic blood pressure after an isometric hand-grip exercise (maintenance of 30% of the maximal hand grip strength for 3–4 min) using a hand dynamometer was recorded [28,30,31]. A rise in diastolic blood pressure of <16 mm Hg was considered abnormal [28,31]. The heart rate variability (heart rate during expiration: heart rate during inspiration) was recorded by electrocardiogram and a ratio of ≤1.1 was considered abnormal [28,32,33]. Abnormality in any one of the above-mentioned tests was regarded as cardiac autonomic neuropathy [28,34,35].

### 2.5. Sample Size and Statistical Analyses

All eligible patients who attended the health consultation clinic were included; hence, a complete enumeration method was followed, and no sampling technique was used. The categorical variables were represented by proportions, while the continuous variables were represented by the mean and standard deviation (after ascertaining the normal distribution of the variables by the Kolmogorov–Smirnov test). Descriptive statistics were used to represent the demographic variables, details of medication intake, anticholinergic burden, number of potential inappropriate medications, and parameters reflecting the cardiac autonomic function. Associations between categorical variables were checked by the unpaired Student’s *t*-test (continuous variables) or the Pearson’s chi-square test (categorical variables) between elderly patients who were on polypharmacy and those who were not. Binary logistic regression analysis was performed to find out the association of polypharmacy and total anticholinergic burden with cardiac autonomic neuropathy in terms of adjusted odds ratio (aOR) with 95% confidence intervals (CI) taking demographic variables and status of disease control as covariates. The model fitness was checked by the Omnibus chi-square test and the Hosmer–Lemeshow test. The level of significance was set at 5%. All analyses were conducted in SPSS version 21.0 (IBM, Armonk, NY, USA).

### 2.6. Ethics Approval

This study was initiated after obtaining appropriate ethical approval. The study was conducted according to the principles of the Declaration of Helsinki, 1964, and its subsequent amendments, and the Indian Council of Medical Research National Ethical Guidelines for Biomedical and Health Research Involving Human Participants, 2017. Because of the use of record-based data without any patient identifier, the requirement of informed consent from the patients was waived.

## 3. Results

A total of 321 patients attended the health consultation clinic; of them, 305 fulfilled the eligibility criteria and were considered in the study. The remaining 16 patients either had cancer or were unfit to undergo cardiac autonomic function tests and were excluded from the analysis. The mean age of the patients was 68.9 ± 3.4 years and 201 patients (65.9%) were male. The majority of the patients had hypertension (81.9%) and type 2 diabetes mellitus (79%) as comorbidities. Two, three, and more than three comorbidities were present in 277 (90.8%), 219 (71.8%), and 103 (33.8%) patients, respectively. A total of 81 patients (26.6%) were on polypharmacy. The most common medicines prescribed to the patients were metformin (221, 72.4%). The distribution of the background characteristics of all elderly patients, those who were on polypharmacy, and those who were not are enumerated in Table 1. There was no significant difference in the distribution of most of the background characteristics between elderly patients who were on polypharmacy and those who were not, indicating the comparability between the two groups.

Ninety-eight potential inappropriate medications (STOPP/START criteria) among the 81 patients who were on polypharmacy were detected in 42 patients (51.8%). The anticholinergic burden was significantly higher among patients who were on polypharmacy than those who were not (8.1 ± 2.3 vs. 2.3 ± 0.9; *p* = 0.03). The proportion of patients with cardiac autonomic neuropathy was significantly higher among those who were on polypharmacy as compared to those who were not (56.8% vs. 44.6%; *p* = 0.01). The results of the individual parameters of the cardiac autonomic function test are enumerated in Table 2. A total of 163 (53.4%) patients had a total anticholinergic burden of > 3. The correlation between the total number of medicines taken and the total anticholinergic burden in the study population is shown in Figure 1. The binary logistic regression model was found to be a good fit as evidenced by the significant Omnibus chi-square test and the non-significant Hosmer Lemeshow statistics. It was found that the presence of polypharmacy and a total anticholinergic burden of >3 was associated with higher odds of having cardiac autonomic neuropathy (aOR, 2.66; 95% CI, 0.91–3.98 and aOR, 2.51; 95% CI, 0.99–3.52, respectively).

## 4. Discussion

In this study, we found that the anticholinergic burden and the proportion of patients with cardiac autonomic neuropathy were significantly higher among patients who were on polypharmacy than those who were not. Further, the presence of polypharmacy and a total anticholinergic burden of >3 was significantly associated with higher odds of having cardiac autonomic neuropathy.

Aging gradually affects the physiological function of every major organ system. This transition invariably leads to the development of multiple chronic diseases, such as hypertension, type 2 diabetes mellitus, coronary artery disease, etc. Polypharmacy is very common among elderly patients, because of these multiple chronic conditions. It is prevalent in hospital settings, ambulatory care settings, as well as nursing home/old age care home settings [1]. Inappropriate polypharmacy increases the risk of adverse drug reactions, harmful drug–drug interactions, and drug–disease interactions. Medication non-adherence and increased healthcare costs are associated with adverse consequences [36]. Continuous research on the effect of polypharmacy in elderly patients is important to generate more physiological evidence. The elderly patients who participated in this study had a controlled chronic disease status; yet, we found that polypharmacy was associated with cardiac autonomic neuropathy irrespective of their disease control status. We herein highlighted the viewpoint of the World Health Organization [1] and the National Institutes of Health—National Institute on Ageing [37] on Clinical Pharmacologist-led routine medication review, de-prescribing, and medication reconciliation exercise among elderly patients who are on polypharmacy [38]. 

Anticholinergic burden is defined as the “accumulation of higher levels of exposure due to one or more anticholinergic medications and the attendant increased risk of medication-related adverse effects” [39]. Polypharmacy and anticholinergic burden are both common problems in elderly individuals [26]. The anticholinergic burden of medicines puts elderly patients at increased risk of negative clinical outcomes, such as falls, delirium, hospitalization, functional decline, and negative impact on cognitive functions [18]. In our study, the overall proportion of patients with cardiac autonomic neuropathy was high, irrespective of the presence of polypharmacy. This can be attributed to the inclusion of a significant proportion (81.9%) of patients with hypertension in our study. There is an age-related physiological decline in cardiac autonomic function [40,41].

In general, elderly patients are at particular risk of orthostatic hypotension and syncope because of altered baroreceptor responsiveness, polypharmacy, and the increased risk of volume depletion [42]. As mentioned, there is a high rate of polypharmacy in elderly individuals with an anticholinergic medication burden identified [43]. Further, pharmacological therapy is one of the main causes of orthostatic blood pressure impairment, leading to iatrogenic orthostatic hypotension [21]. Elderly patients and those with polypharmacy are particularly at risk of orthostatic hypotension [19,20]. Fall risk is associated with the use of polypharmacy, but only when at least one established fall risk-increasing medication is a part of the patient’s daily medication regimen [44]. Previous studies have shown that cardiac autonomic neuropathy among elderly individuals is associated with frailty, functional decline, and poor prognosis [22,45,46,47]. Thus, polypharmacy in elderly individuals may continue to influence daily heart rate variability and increase the risk of pathogenesis in the future [23]. Hence, it is not only important to control the onset of chronic diseases with medications, but it is also necessary to consider the number of medications and their effects on elderly patients. Physicians should routinely monitor the cardiovascular autonomic function among elderly patients on polypharmacy. If there is any abnormality, immediate measures should be taken to mitigate the risks of adverse consequences. One such measure can be medication de-prescribing [6,48].

This study was based on a record register of a Clinical Pharmacology health consultation clinic and focused on community-dwelling free-living elderly patients. There is a dearth of studies on the association between polypharmacy and cardiovascular autonomic function among the elderly population in India. This is the strength of the present study. However, there are some limitations. First, the sample size was comparatively small in this record-based observational study. This was a single-center study from an urban municipality area of Kolkata, India. As mentioned, the prevalence of polypharmacy varies widely from 2–93% across various Indian states depending on several factors, such as the sample size of the study; the definition of polypharmacy used; and differences in socioeconomic conditions, risk factors, and quality of healthcare services [4,5]. However, we did not primarily aim to find the prevalence of polypharmacy in the study area. Second, most of the elderly patients had controlled chronic disease status, with no need for nursing care. In the real-world scenario, there are more confounding factors that affect cardiac autonomic functions in elderly patients. Third, advanced techniques [49,50] for the evaluation of cardiovascular autonomic function and its neuronal regulation were not used. Finally, the effect of the duration of polypharmacy on cardiac autonomic neuropathy was not considered. Notwithstanding these limitations, it was documented that polypharmacy was significantly associated with cardiac autonomic neuropathy among community-dwelling elderly patients. The results of the present study can be considered as preliminary findings which should be confirmed by further studies utilizing a robust methodology in a larger population. 

## 5. Conclusions

We found that the anticholinergic burden and the proportion of patients with cardiac autonomic neuropathy were significantly higher among patients who were on polypharmacy than those who were not. The presence of polypharmacy and a total anticholinergic burden of >3 was associated with higher odds of having cardiac autonomic neuropathy among community-dwelling elderly patients.

## Figures and Tables

**Figure 1 geriatrics-07-00136-f001:**
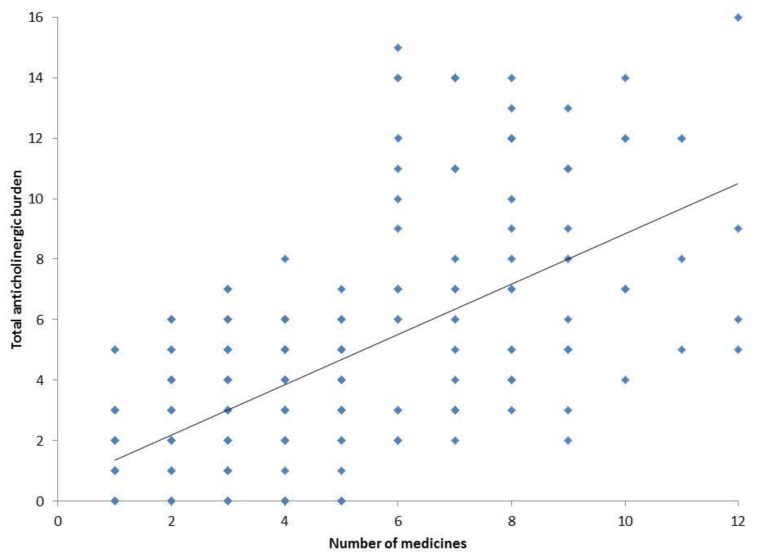
Correlation between the total number of medicines taken and total anticholinergic burden in the study population. Each dot represents the data of individual patients (*n* = 305).

**Table 1 geriatrics-07-00136-t001:** The distribution of the background characteristics of all elderly patients, those who were on polypharmacy, and those who were not are enumerated in Table 1.

Parameters	All Elderly Patients (*n* = 305)	Elderly Patients Who Were on Polypharmacy (*n* = 81)	Elderly Patients Who Were Not on Polypharmacy (*n* = 224)
Demographic characteristics			
Age (years)	68.9 ± 3.4	69.2 ± 3.5	68.8 ± 3.2
Male (%)	201 (65.9)	52 (64.2)	149 (66.5)
Distribution of comorbidities *			
Hypertension	250 (81.9)	64 (79.0)	186 (83.0)
Type 2 diabetes mellitus	241 (79.0)	65 (80.2)	176 (78.6)
Dyspepsia and chronic gastritis	239 (78.4)	73 (90.1)	166 (74.1) ^†^
Coronary artery disease	213 (69.8)	56 (69.1)	157 (70.1)
Chronic pulmonary obstructive disease	147 (48.2)	40 (49.4)	107 (47.8)
Osteoarthritis	88 (28.8)	30 (37.0)	58 (25.9) ^†^
Benign prostatic hyperplasia	77 (25.2)	10 (12.3)	67 (29.9) ^†^
Hypothyroidism	77 (25.2)	19 (23.5)	58 (25.9)
Anxiety and depression	76 (24.9)	32 (39.5)	44 (19.6)
Urinary tract infection	56 (18.4)	18 (22.2)	38 (16.9) ^†^
Cerebrovascular disease	32 (10.5)	8 (9.9)	24 (10.7)
Distribution of medication intake *			
Metformin	221 (72.4)	56 (69.1)	165 (73.7)
Pantoprazole	220 (72.1)	76 (93.8)	144 (64.3) ^†^
Losartan	201 (65.9)	56 (69.1)	145 (64.7)
Amlodipine	199 (65.2)	54 (66.7)	145 (64.7)
Glimepiride	120 (39.3)	40 (49.4)	80 (35.7) ^†^

The results are expressed as the mean ± standard deviation or number (%). * Multiple responses were taken. ^†^ *p* < 0.05 between elderly patients who were on polypharmacy and those who were not.

**Table 2 geriatrics-07-00136-t002:** Comparison of the anticholinergic burden and proportion of patients with cardiac autonomic dysfunction between elderly patients who were on polypharmacy and those who were not (*n* = 305).

Parameters	Elderly Patients Who Were on Polypharmacy (*n* = 81)	Elderly Patients Who Were Not on Polypharmacy (*n* = 224)	*p*-Value
Anticholinergic burden	3.0 ± 0.3	1.6 ± 0.1	0.03
Cardiac autonomic dysfunction	46 (56.8%)	100 (44.6%)	0.01
Abnormal Valsalva ratio	35 (43.2%)	81 (36.2%)	0.02
Orthostatic hypotension	32 (39.5%)	72 (32.1%)	0.01
Abnormal hand-grip exercise test	25 (30.9%)	62 (27.7%)	0.09
Abnormal expiration: inspiration time ratio	28 (34.6%)	65 (29.0%)	0.07

The results are expressed as the mean ± standard deviation or number (%).

## Data Availability

The datasets generated during and/or analyzed during the current study are available from the corresponding author upon reasonable request.

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
