# Peer review of "Association between Polypharmacy and Cardiovascular Autonomic Function among Elderly Patients in an Urban Municipality Area of Kolkata, India: A Record-Based Cross-Sectional Study"

_geriatrics, 2022, doi:10.3390/geriatrics7060136_

Round 1
Reviewer 1 Report
This is an interesting and important study to show polypharmacy was significantly associated with cardiac autonomic neuropathy among community-dwelling elderly patients in a record-based cross-sectional study.
The title of this report “Influence of polypharmacy on cardiovascular autonomic function among” seems to be inappropriate. The association between them is found but influence of polypharmacy is not studied.
It is necessary to show a new table showing a comparison of background characteristics between polypharmacy and non-polypharmacy groups. Effects from covariates are unclear. Multivariate analysis is required.
Author Response
We thank the editor and the reviewers for taking the time and effort to review the manuscript. We appreciate the valuable comments received; these will certainly improve the quality of the manuscript. We have responded to each comment below.
Reviewer 1
Comment: This is an interesting and important study to show polypharmacy was significantly associated with cardiac autonomic neuropathy among community-dwelling elderly patients in a record-based cross-sectional study.
Response: Thank you for the encouraging comments.
Comment: The title of this report “Influence of polypharmacy on cardiovascular autonomic function among” seems to be inappropriate. The association between them is found but influence of polypharmacy is not studied.
Response: Thank you for this suggestion. We have changed the title of the manuscript to “Association between polypharmacy and cardiovascular autonomic function among elderly patients in an urban municipality area of Kolkata, India: A record-based cross-sectional study”.
Comment: It is necessary to show a new table showing a comparison of background characteristics between polypharmacy and non-polypharmacy groups. Effects from covariates are unclear. Multivariate analysis is required.
Response: We have modified Table 1. The distribution of the background characteristics of all elderly patients, those who were on polypharmacy, and those who were not are enumerated in Table 1. There was no significant difference in the distribution of most of the background characteristics between elderly patients who were on polypharmacy and who were not, which indicates the comparability between the two groups. Multivariate analysis i.e. binary logistic regression analysis was performed to find out the association between polypharmacy and total anticholinergic burden with cardiac autonomic neuropathy. The dependent variable was the presence of cardiac autonomic neuropathy. The binary logistic regression model was found to be a good fit as evidenced by the significant Omnibus chi-square test and the non-significant Hosmer Lemeshow statistics. It was found that the presence of polypharmacy and a total anticholinergic burden of >3 was associated with higher odds of having cardiac autonomic neuropathy (aOR, 2.66; 95% CI, 0.91–3.98 and aOR, 2.51; 95% CI, 0.99–3.52, respectively).
Reviewer 2 Report
“Influence of polypharmacy on cardiovascular autonomic function among elderly patients: A record-based cross-sectional 3 study” is a record-based cross-sectional study conducted in an urban municipality area of Kolkata, India, considering 305 patients (age, 68.9±3.4; 65.9% male), of whom 81 (26.6%) were on polypharmacy.
The anticholinergic burden and the proportion of patients with cardiac autonomic neuropathy were significantly higher among patients on polypharmacy than those without. Of note, inappropriate medications (STOPP/START criteria) were found in 51.8%. Two, three, and more than three 141 comorbidities were present in 277 (90.8%), 219 (71.8%), and 103 (33.8%) patients, respectively.
The authors conclude that polypharmacy was significantly associated with cardiac autonomic neuropathy among community-dwelling elderly patients.
It is an interesting paper dealing with an emerging problem in the older persons, such a problem is well discussed in the introduction.
Major concerns:
1) The main issue is related to the fact that is not possible to ascertain whether the cardiac autonomic imbalance is due to polypharmacy or to the presence of diseases, known to cause cardiovascular autonomic dysregulation. The fact that in the polypharmacy group 52% of patients were on potentially inappropriate treatment may support the authors’ conclusions, it is reasonable that the effect on autonomic parameters may be related, at least in part, to polypharmacy. However, data presented are not convincing. I suggest to consider and show presence of diseases in the two groups (the authors mention the percentage of each disease in the whole cohort) and weight their influences on the autonomic parameters. Ideally, the 2 groups should be perfectly balanced for distribution of pathologies. If this were not the case, the results would be certainly weaker. According to the results, the authors should discuss appropriately and clarify possible limitations (the number of enrolled patients is probably small).
2) The cardiac autonomic function was evaluated by means of clinical indices and time-domain parameters of HRV. More sophisticated methods are available for the evaluation of the cardiac and cardiovascular neural regulation (see for example the following studies: - Eur Heart J. 1996 Mar;17(3):354-81; - doi: 10.1038/s41440-018-0056-y. - doi: 10.1007/s00399-021-00780-5. Epub 2021 Jul 8. PMID: 34236476., - doi: 10.1016/j.ejim.2004.06.016. PMID: 15733815. - doi: 10.1007/s11517-017-1765-0) this point should be mentioned as a limitation. The results aìof the present study can be considered as preliminary findings which should be confirmed by further studies utilizing sound methodology on a larger population.
Minors:
1) The prevalence of polypharmacy very different around the world, as stated in the introduction, 40–50% in high-income countries, from 2–93% in India, a wide range probably due to differences between urban and non-urban areas, data of the present manuscript are based on a single center study in an urban area of India. This should be clear in the title and these wide ranges should be discussed in the paper.
2) any difference between men and women? As known, sex/gender differences should be checked in each stdy.
Author Response
Comment: “Influence of polypharmacy on cardiovascular autonomic function among elderly patients: A record-based cross-sectional 3 study” is a record-based cross-sectional study conducted in an urban municipality area of Kolkata, India, considering 305 patients (age, 68.9±3.4; 65.9% male), of whom 81 (26.6%) were on polypharmacy. The anticholinergic burden and the proportion of patients with cardiac autonomic neuropathy were significantly higher among patients on polypharmacy than those without. Of note, inappropriate medications (STOPP/START criteria) were found in 51.8%. Two, three, and more than three 141 comorbidities were present in 277 (90.8%), 219 (71.8%), and 103 (33.8%) patients, respectively. The authors conclude that polypharmacy was significantly associated with cardiac autonomic neuropathy among community-dwelling elderly patients. It is an interesting paper dealing with an emerging problem in the older persons, such a problem is well discussed in the introduction.
Response: Thank you for the encouraging comments.
Comment: The main issue is related to the fact that is not possible to ascertain whether the cardiac autonomic imbalance is due to polypharmacy or to the presence of diseases, known to cause cardiovascular autonomic dysregulation. The fact that in the polypharmacy group 52% of patients were on potentially inappropriate treatment may support the authors’ conclusions, it is reasonable that the effect on autonomic parameters may be related, at least in part, to polypharmacy. However, data presented are not convincing. I suggest to consider and show presence of diseases in the two groups (the authors mention the percentage of each disease in the whole cohort) and weight their influences on the autonomic parameters. Ideally, the 2 groups should be perfectly balanced for distribution of pathologies. If this were not the case, the results would be certainly weaker. According to the results, the authors should discuss appropriately and clarify possible limitations (the number of enrolled patients is probably small).
Response: Thank you for this suggestion. We agree to this issue. The elderly patients who participated in this study had a controlled chronic disease status. In the binary logistic regression analysis, demographic variables and the status of disease control were taken as covariates. In the polypharmacy group, 52% of the patients were on potentially inappropriate treatment. This supports our finding that polypharmacy was significantly associated with cardiac autonomic neuropathy among community-dwelling elderly patients. We have modified Table 1. The distribution of the background characteristics of all elderly patients, those who were on polypharmacy, and those who were not are enumerated in Table 1. There was no significant difference in the distribution of most of the background characteristics between elderly patients who were on polypharmacy and who were not. We have mentioned in the limitation that the sample size was comparatively small.
Comment: The cardiac autonomic function was evaluated by means of clinical indices and time-domain parameters of HRV. More sophisticated methods are available for the evaluation of the cardiac and cardiovascular neural regulation (see for example the following studies: - Eur Heart J. 1996 Mar;17(3):354-81; - doi: 10.1038/s41440-018-0056-y. - doi: 10.1007/s00399-021-00780-5. Epub 2021 Jul 8. PMID: 34236476., - doi: 10.1016/j.ejim.2004.06.016. PMID: 15733815. - doi: 10.1007/s11517-017-1765-0) this point should be mentioned as a limitation. The results of the present study can be considered as preliminary findings which should be confirmed by further studies utilizing sound methodology on a larger population.
Response: We have added the following sentences under limitations: “Advanced techniques [49, 50] for the evaluation of cardiovascular autonomic function and its neuronal regulation were not used. The results of the present study can be considered as preliminary findings which should be confirmed by further studies utilizing a robust methodology in a larger population.”
Comment: The prevalence of polypharmacy very different around the world, as stated in the introduction, 40–50% in high-income countries, from 2–93% in India, a wide range probably due to differences between urban and non-urban areas, data of the present manuscript are based on a single center study in an urban area of India. This should be clear in the title and these wide ranges should be discussed in the paper.
Response: Thank you for this suggestion. We have changed the title of the manuscript to “Association between polypharmacy and cardiovascular autonomic function among elderly patients in an urban municipality area of Kolkata, India: A record-based cross-sectional study”. We have also discussed the variation in the prevalence of polypharmacy across India under limitations, “As mentioned, the prevalence of polypharmacy varies widely from 2–93% across various Indian states depending on several factors, such as sample size of the study; definition of polypharmacy used; and differences in socioeconomic conditions, risk factors, and quality of healthcare services across various Indian states [4, 5]. However, we did not primarily aim in finding the prevalence of polypharmacy in the study area.”
Comment: Any difference between men and women? As known, sex/gender differences should be checked in each study.
Response: We have modified Table 1. The distribution of the background characteristics of all elderly patients, those who were on polypharmacy, and those who were not are enumerated in Table 1. There was no significant difference in the distribution of most of the background characteristics between elderly patients who were on polypharmacy and who were not.
Round 2
Reviewer 1 Report
The article shows that polypharmacy was significantly associated with cardiac autonomic 247 neuropathy among community-dwelling elderly patients.
The manuscript is sufficiently modified and well witten.
Reviewer 2 Report
Thank you for your replies.